# Research and Development Journey and Future Trends of Hollow Fiber Membranes for Purification Applications (1970–2020): A Bibliometric Analysis

**DOI:** 10.3390/membranes11080600

**Published:** 2021-08-07

**Authors:** Muhammad Ayub, Mohd Hafiz Dzarfan Othman, Siti Hamimah Sheikh Abdul Kadir, Adnan Ali, Imran Ullah Khan, Mohd Zamri Mohd Yusop, Takeshi Matsuura, Ahmad Fauzi Ismail, Mukhlis A. Rahman, Juhana Jaafar

**Affiliations:** 1Advanced Membrane Technology Research Centre (AMTEC), School of Chemical and Energy Engineering, Universiti Teknologi Malaysia (UTM), Johor Bahru 81310, Johor, Malaysia; ayub1977@graduate.utm.my (M.A.); zamriyusop@utm.my (M.Z.M.Y.); afauzi@utm.my (A.F.I.); mukhlis@petroleum.utm.my (M.A.R.); juhana@petroleum.utm.my (J.J.); 2Institute of Pathology, Laboratory and Forensic Medicine (I-PPerForM), Universiti Teknologi Mara (UiTM), Cawangan Selangor, Kampus Sungai Buloh, Jalan Hospital, Sungai Buloh 47000, Selangor, Malaysia; 3Azman Hashim International Business School (AHIBS), Universiti Teknologi Malaysia (UTM), Johor Bahru 81310, Johor, Malaysia; ali.adnan@graduate.utm.my; 4Department of Management Sciences, Shaheed Benazir Bhutto University, Sheringal, Dir Upper 18050, Khyber Pakhtunkkhwa, Pakistan; 5Department of Chemical and Energy Engineering, Pak-Austria Fachhochschule, Institute of Applied Sciences & Technology (PAF:IAST), Khanpur Road, Mang, Haripur 22650, Pakistan; imran.khan@fcm3.paf-iast.edu.pk; 6Department of Chemical and Biological Engineering, University of Ottawa, Ottawa, ON K1N 6N5, Canada; matsuura@uottawa.ca

**Keywords:** hollow fiber membrane, bibliometric analysis, VOSviewer, gas separation, water treatment, future trends

## Abstract

Hollow fiber membrane (HFM) technology has received significant attention due to its broad range separation and purification applications in the industry. In the current study, we applied bibliometric analysis to evaluate the global research trends on key applications of HFMs by evaluating the global publication outputs. Results obtained from 5626 published articles (1970–2020) from the Scopus database were further manipulated using VOSviewer software through cartography analysis. The study emphasizes the performance of most influential annual publications covering mainstream journals, leading countries, institutions, leading authors and author’s keywords, as well as future research trends. The study found that 62% of the global HFM publications were contributed by China, USA, Singapore, Japan and Malaysia, followed by 77 other countries. This study will stimulate the researchers by showing the future-minded research directions when they select new research areas, particularly in those related to water treatment, biomedical and gas separation applications of HFM.

## 1. Introduction

Membrane separation processes using hollow fibers are one of the technologies that have experienced a rapid growth during the last two decades. Numerous applications of hollow fiber membranes (HFM) having great commercial interest have been identified with respect to separation and purification needs, mainly in household water applications, the industry and the biomedical field. The preparation of potable and high-quality water for the household, electronics and pharmaceutical industry, wastewater treatment, liquid phase separations, gas separations for industrial applications, removal of volatile organic compounds (VOCs) from water, hemodialyzers and the controlled release of drugs are only a few of those examples [1,2,3,4,5]. HFMs are preferred due to distinct advantages over flat or tubular membranes, such as a large effective surface-to-volume ratio, conceptual simplicity, minimal fouling, easy incorporation into flow streams and broad availability [6]. HMF can be applied in ultrafiltration, membrane contactors, pervaporation, microfiltration, reverse osmosis, forward osmosis (FO), pressure retarded osmosis (PRO) and many other liquid/liquid or liquid/solid separation applications [7,8]. Membranologists are currently seeking suitable membranes as a primary separation barrier that can be effective for industrial applications. Sustainable processes and environment-friendly preparation techniques have recently been developed in making high-performance hollow fiber membranes [5].

For polymeric hollow fiber membranes (PHFM), spinning parameters are the crucial factors that must be controlled during the preparation of membranes. Hollow fiber spinning is a tricky physical process and generally involves the following four steps: (i) solution formulation, (ii) extrusion, (iii) coagulation and (iv) treatment of coagulated fiber. The most common techniques used to prepare polymeric membranes are the phase inversion process, thermally induced phase separation (TIPS) and diffusion-induced phase separation (DIPS) [5,9]. It appears in the literature that poly (vinylidene fluoride) (PVDF) hollow fiber has attracted much more attention in comparison with other polymers due to its noticeable advantages, such as its stability against vigorous chemicals that are used for membrane cleaning and considerable thermal resistance. During hollow fiber spinning, many variables are involved, and the control of these variables may enable the fabrication of membranes with the desired morphologies and physical properties. It has been found in the literature that mixed matrix hollow fiber membranes have a great potential for the gas separations [4,10,11,12].

Besides PHFMs, ceramic HFMs (CHFMs) have been developed based on robust ceramic material support to be used as alternatives to PHFMs because of their higher chemical, thermal and mechanical stability. Many types of ceramic materials, such as silica, alumina, and perovskites, are used in the preparation of CHFM, which can be more favorable under harsh application conditions, such as high temperatures (>100 C), a wider pH range and corrosive and fouling environments. Many applications of CHFM have been explored based on various modifications in CHFM preparation. Water desalination using membrane distillation/ultrafiltration/microfiltration, O_2_ production, H_2_ production, dehydration of organic solvents and solid oxide fuel cells are some of the emerging applications of CHFM [9,13,14,15].

Previous studies suggested that HFM plays an important role in gas separation technology, water treatment and some biomedical applications, including artificial kidney, artificial lungs, blood oxygenators and artificial liver, due to its high separation areas and selectivity [3,16,17]. Due to the exponential growth of HFM research and development and increasing number of publications, a critical analysis of its past, current and future study is urgently needed. A quantitative and qualitative assessment is required to provide scientific guidelines for a better understanding of what role HFMs are playing in key areas, such as wastewater treatment, gas separation and biomedical applications, and to know what current and future trends in these research areas are. The studying of bibliometric analysis is a systematic approach for comprehending the global research trends from the outputs of the academic literature database, in a certain research area. As compared to the orthodox literature reviews, bibliometric analysis is free from any subjectivity biases, keeping the importance and sensitivity of the research area and its development over time. Moreover, a new researcher cannot go through a bulk of literature to understand the developments, and therefore a scientific approach, such as a bibliometric analysis, can quench the thirst by offering a bird’s-eye view on that area of interest, including the current trends and future directions. It is a meta-analytical tool that demonstrates interconnections among research articles and topics based on co-citation analysis [18,19] and analyses how often an article is cited by other articles, indicating key research streams for a specific topic [20]. It enables researchers and authors to gain a clear view of the structure of the given field [21]. The software, visualization of similarities (VOSviewer), highlights the most-cited articles and provides sketch visualization graphs of citations as well [22].

This kind of approach distinguishes a paper, based on bibliometric analysis, from a review-based paper, which is primarily projected to discuss the latest progress, challenges, and future directions of a certain topic. It is based on the literature, using the various characteristics of the literature, by applying mathematical and statistical methods to collect, organize and process information that can be used for the development of research trends of a professional discipline during a given period [23]. Although there has been a growing interest in the R&D of HFM technology, including its new applications, no study has yet been dedicated to measuring and analyzing HFM related scientific publications from a global perspective. Miranda et al [24] presented an extensive bibliometric survey regarding the use of membranes for gas separations in petrochemical processes from 1960 to 2019 using electronic search tools, such as Google Scholar, Google Patents, USPTO and EspaceNet. Dai et al. [25] also performed a bibliometric study on research progress in the area of membranes for water treatment from 1985 to 2013 using the online ISI Web of Science Core Collection database. Other than membranes, a lot of topics have been analyzed through bibliometric analysis, which shows its importance from the future prospective [26,27,28].

Present work used the Scopus database and VOSviewer to conduct a bibliometric analysis covering the entire period of 1970–2020 for evaluating the networks among highly cited articles, including their authors, countries, institutions, and journals. This study also examined which journals, articles, authors, institutions, and countries contributed most to HFM research, using co-citation bibliometric and cartography analysis along with content analysis of the literature. In addition, by using VOSviewer software, frequently occurring keywords and their link strengths with other keywords in HFM research streams were also found. This paper is an attempt to explore and investigate for the new horizons in the development and applications of HFM-related scientific studies using the latest techniques. To the best of our knowledge, this may be the first comprehensive analysis of its type to know the past, present and future research trends of HFMs and their applications. The achievements of this paper will open new avenues for other membrane researchers to identify the past research advancements and clarify the future research directions in HFMs.

## 2. Materials and Methods

### 2.1. Data Source and Methodology

According to Scopus database, the global literature about hollow fiber membranes (HFMs) published on the oldest publication year i.e., 1970 to the most recent publication year 2020 were scanned online and included in the current work. Scopus is a source-neutral abstract and citation database provided by Elsevier, a Netherlands-based information and analytics company. It is recognized as a huge multidisciplinary abstract and citation database consisting of 75M+ records of peer-reviewed literature, from a wide range of subjects [29]. Thus, the use of Scopus enables us to cover those topics which may not be offered in Web of Science or other databases [30]. After various trials for selecting and organizing keywords, the search for finalized key strings was conducted on 3rd January 2021. The search terms “hollow fiber membrane” OR “dual-layer hollow fiber membrane” were employed to identify the closest matching publication in which the above keywords were included in the title and abstract. A total of 7363 records were collected. After excluding the year 2021 and limiting the search to English language and the document types to only journal articles, a total of 5668 papers were retrieved. The query string used was:

TITLE-ABS (“hollow fiber membrane”) OR TITLE-ABS (“dual-layer hollow fiber membrane”) AND (EXCLUDE (PUBYEAR, 2021)) AND (LIMIT-TO (DOCTYPE, “ar”)) AND (LIMIT-TO (LANGUAGE, “English”)) AND (LIMIT-TO (SRCTYPE, “j”)).

Among these 5668 articles, 32 more papers were excluded because these were identified as review papers by going through all titles, abstracts and, if needed, through full texts of 5668 articles. Five more papers having no author and five papers with duplicate titles (Scopus limitation) were also traced and excluded from the total count. Finally, after excluding the EIDs (electronic identification, a Scopus unique article identifier number) of these 42 irrelevant papers in the Scopus search string, a total of 5626 articles were further subjected to scientometric and systematic analysis. Field code “AFFILCOUNTRY” was used to retrieve the data for single-country publication (SCP) for limiting the search result to a specific country. These search results (5626 articles) were analyzed based on publication year, author, country/territory, institution, subject area, and document type. For ranking and further analysis, bibliometric indicators (total publications, total citations, and document h-index) were used. By scrutinizing the output trends in the major HFM applications, top three terms were selected as a secondary theme, based on the most occurring keywords analysis in the software, which were found to be gas separation, water treatment and biomedical applications. The search string for each of above given secondary theme applications was run separately in Scopus database, in combination with the central theme (HFM). These search results were analyzed based on publication output per year.

### 2.2. Analytical Tools and Procedure

Bibliometrics/scientometrics is a set of methods used to analyze academic literature quantitatively in a systematic way [31,32]. VOSviewer is a software tool developed by Centre for Science and Technology Studies, Leiden University, Leiden, The Netherlands, for constructing and visualizing bibliometric maps [33]. The program is freely available (www.vosviewer.com accessed on 7 August 2021) for everyone interested in bibliometric analysis. VOSviewer can be used to construct maps of authors or journals based on co-citation data or to construct maps of keywords based on co-occurrence data. This program offers its viewer the method to examine bibliometric maps in full detail. VOSviewer can also display a map in multiple formats, each emphasizing a different aspect of the map. Maps created by VOSviewer use the term items, and in the current study, the items are the objects of interest, namely the countries or author keywords. In the present work, version 1.6.15 of this software was used to analyze the results obtained from Scopus database [33]. It was found that a total of 355 studies used VOSviewer for bibliometric analysis using Scopus database, of which 78% were from the past two years (2019–2020) and 56% only in the year 2020, which is the highest figure as compared to CiteSpce (220 documents), HistCite (51 documents) or any other software, and it also shows the popularity and significance of this software for comprehensive scientometric analysis. VOSviewer has been used successfully in various projects carried out by the Centre for Science and Technology Studies [34].

A total of 5626 HFM articles that meet the requirements, contained year of publication, journal, title, authors’ names, author ID, affiliation, citation count, EID and abstract, which were exported into CSV (comma-separated values) file format on 3rd January 2021. After removing the EIDs for the review articles, duplicates, missing authors name and IDs, the refined results were exported to MS Excel for presenting the annual publications and cumulative publications trend per year since 1970 to 2020, as shown in Figure 1. This information is valuable for scientometric analysis. The CSV files were also imported to VOSviewer to analyze the co-authorship, co-occurrence, bibliographic coupling, co-citation and other themes. These outputs allowed us to explore the research streams of HFM articles in various applications.

Two standard weight attributes, which are defined as “links attribute” and “link strength attribute”, are applied. The links and link strength attributes are represented by a positive numerical value. A higher value shows stronger link and vice versa. In a bibliographic coupling analysis, connections among different authors are determined by assessing the degree to which they cite the same research publications i.e., the higher the citations the stronger the link strength and vice versa [34]. The link strength between different countries in co-authorship analysis shows the number of co-authored publications between these countries, whereas the total link strength (TLS) indicates the overall co-authorship links of a given country with other countries. The case of co-occurrence analysis for author keywords is the same. For example, a keyword, such as “graphene oxide (GO)”, has 30 links, which means it is connected to 30 other keywords, and higher link strength value of a keyword with GO shows the stronger research interest of that area with GO. Detailed features of VOSviewer can be found in the user manual [33]. Furthermore, VOSviewer allocates research articles, authors, countries and keywords into various clusters by identifying the similarities in topic, theme or research field [35]. Therefore, HFM research streams were confirmed with cartography analysis. In VOSviewer, minimum occurrences of a keyword to be analyzed was set to 5. Overlay network and density visualization mode were used to view the average publication year, number of occurrences and link strength of the keywords. The color of a keyword in overlay visualization mode indicates the average publication year of the documents in which a keyword occurs.

## 3. Results and Discussions

### 3.1. Scientometric Analysis of Publication Output

From the Scopus search, 15 document types were found among the total 7363 publications during the 51-year study period, and the most frequent document type was articles (6392), which were responsible for 87% of the total publications. However, after screening, as already discussed in Section 2.1, 5626 original scientific articles were taken for further analysis as shown in Figure 1. The first article on HFMs indexed in the Scopus database was published in 1970 in the journal of *Environmental Science & Technology* under “ACS Publications”, titled as “ Decarbonation and Deaeration of Water by use of Selective Hollow Fibers” [36]. However, during the period of 15 years (1970–1984), there were only 91 publications on HFM research, which shows the little interest and awareness of HFM for its applicability in different areas of human development. However, it was much explored by researchers in the next 15 years (1985–1999), which is apparent by the increasing trend in HFM publications from 91 to 906 (995% increase). Research on HFM and its versatile applications increased continually and rocketed in the past two decades (2001–2020) with a total of 5626 research articles, achieving the highest figure in the year 2018 (375 articles), as shown in Figure 1. This variation trend of research papers reflected the development process of HFM applications specifically for gas separation and water treatment technology [3,37]. Factors such as increasing problems of water pollution and water scarcity with an increasing global population contributed to the development of new methods for water purification. Furthermore, the improvement in people’s living standards for using water of high quality also diverted the attention of researchers to finding new membrane solutions of water treatment. This rising number of publications suggests a clear research focus and importance of HFM and its applications in industrial separations (filtration of drinking water), gas separations, membrane contactors, the biomedical industry (artificial kidney and oxygenators), ultrafiltration, microfiltration and HFM bioreactors [38,39]. Therefore, it is expected that the annual publication will continue to rise in the coming years. However, as most of these articles are only available on payment, open access options might expand the reading circle and citations.

As for the publishing language, results also showed that all 6216 (including all languages) were published in 14 languages. The English language was at the top with 5626 articles (90.5%), followed by Chinese with 406 articles (6.5%), Japanese with 118 articles (1.9%) and other languages (Korean, German, Russian, French, Spanish, Polish, Portuguese, Arabic, Czech, Italian and Turkish) with 63 articles (1.0%). Moreover, 12 articles (0.2%) were found under undefined language.

### 3.2. Distribution of Articles by Leading Journals and Citations

Table 1 lists the top 10 journals publishing the highest number of publications in the HFM research area. It has been found that *Journal of Membrane Science* published the most articles (1167), which is 20.74% of total HFM publications, followed by *Desalination* with 315 articles (5.60%), and so on, as shown in Table 1. It has been found that these journals belong to four different publishers; Elsevier group is the publisher of six journals, including the top three journals, having a total share of 35.65% in publications. It also confirms that most of the authors publish their work in top ranking journals, specifically Elsevier journals, due to their reliability, quality of work and wider reading circle.

The journal’s impact factor (IF) is an important factor in assessing its value, number of publications and citations and the value of the articles it contains. The IF is a measure of the frequency with which the average article in a journal has been cited in a particular year. It is used to measure the importance or rank of a journal by calculating the citations of the articles the journal published. Although IF is a metric used by Clarivate Analytics for measuring journal impact based on citation data, most authors are more familiar with IF instead of Scopus CiteScore, so we correlated the IF with the number of publications and citations of a journal. The last five years’ IF data of the top ten journals on HFM were retrieved from the Web of Science (WOS) group of Clarivate, as shown in Table 1. It has been found that, overall, a journal with a higher impact factor received more publications and citations with few exceptions, which may be due to strict article selection criteria, number of issues released per year, initial date of registration of journal, publication year of an article, quality of published article, availability of open access option for any article, wider journal scope and many more. As for *Journal of Membrane Science*, it received the highest citations (50,776) although having less five years IF (7.158) than *Desalination* (7.248) and *Chemical Engineering Journal* (9.43). However, on the other hand, it also has the highest number of publications, which is also an important factor for higher citations. It has been found that the most cited article [40], with 468 citations, is published in *AICHE Journal*, which is at number seven in ranking, followed by an article [41] of Journal of Membrane Science, which has 428 citations. Although AICHE Journal has lesser IF, other than the quality of work, the publication year also counts. Here, in this case, the article published in AICHE Journal is 22 years older (1988) than the article of Journal of Membrane Science published in 2010. The information about IF is very important but the journals’ ranking in Table 1 is based on the total publications indexed in the Scopus database.

### 3.3. Co-Authorship for Leading Countries, Top Institutions and International Collaboration

The top 10 most productive countries involved in the research activities related to HFM have been shown in Figure 2 using Google Maps, indicating the location of the country that includes the most productive institution. In Table 2, the 10 most productive countries have been ranked according to the number and percentage of publications with international collaboration (total publication count (TPC), without international collaboration (single country publications (SCP)), name of the institution with its total institute publications (TIP) and QS World University Rankings 2021).

Approximately 39% of the global publications was contributed by China and USA, indicating that these two countries are key players in the HFM research progress. In every aspect, China had a better record than USA, the second productive country. China led by 1260 publications, covering 22.4% of the total global publications, followed by USA with 945 articles. For SCP, it has been shown that China and USA share an almost equal contribution in percentage, whereas the highest SCP (81.9%) has been observed in the case of Japan, which shows that most of the publications by Japan are without international collaborations and have strong intra-country collaboration. The next four countries, identified with a SCP greater than 65%, include USA, China, South Korea and Iran, with strong intra-country partnerships. On the other hand, Australia having 170 publications is interlinked with 24 multiple countries, showing strong inter-country relationships (Figure 3). Other countries, such as the UK and Malaysia, show an almost equal contribution for both inter and intra country collaborations (Table 2). These relationships can be confirmed from Figure 3, showing bibliometric analysis of co-authorship of different countries. The benefits of international cooperation are not only network expansion, knowledge exchange and knowledge sharing, but also an effective strategy for rank-up. For example, in the case of Malaysia, 148 of its publications are international collaborative papers affiliated with 20 countries and ranked fifth in the world competition (Figure 4).

Figure 3 shows the distribution of countries/regions by region. The closer the two countries are to each other in VOSviewer, the deeper their correlation is and the thicker the line of strong co-authorship. The co-authored analysis included all countries and related authors. According to regional division by World Bank, member countries/regions are clustered into seven groups [49]. The highest number of countries per region came from Europe and Central Asia (30), followed by the Middle East and North Africa (16), East Asia and Pacific (13), Latin America and Caribbean (9), Sub-Saharan Africa (4), South Asia (3) and North America (2). Results of co-authorships show that USA was the most affiliated country, linked to 45 countries/territories with 312 instances of co-authorship. The list was followed by France (34 links, 111 co-authorships), UK (32 links, 164 co-authorships), Italy (32 links, 92 co-authorships), China (29 links, 447 co-authorships) and others. China shows a maximum number of co-authorships for its 1269 documents on HFM, with 29 countries as validated by total link strength (447). Several possible factors that contribute to the dynamics of international cooperation may be due to the diversity of research partners, the high proportion of foreign graduate/visiting scholars, and strong research funding. It is also important to have a flexible and stable research policy in order to ensure the sustainability of international cooperation [23]. From Figure 5, it can be found that countries such as USA and Japan have maximum average publications in the year 2006 and earlier, whereas Malaysia, Indonesia, Pakistan, Iran and China show an average publications trend in most recent years.

If we go through the most influential institutions, as referred to in Table 2, the National University of Singapore with the highest TIP (224) is at the top, followed by Tianjin Polytechnic University, China (141) and Universiti Teknologi Malaysia (UTM) (117). The National University of Singapore is at number 11 in the QS world University Ranking 2021, whereas Tianjin Polytechnic University and UTM are at 387 and 187, respectively; so, irrespective of QS ranking, different institutions can have higher or lower number of publications in a specific research area. In addition, there were four universities listed in the top 50 universities based on the QS World University Rankings 2021 [50]. These include Imperial College London (ranking 8th), National University of Singapore (11th), Korea Institute of Energy Research (39th) and UNSW Australia (44th), which reveals that the HFM research area has enough of a share in the top universities in the world. If we go further into the data for institutions given in Table 2, it has been found that three institutions have individual publications greater than 100, which is due to the establishment of membrane research centres in these institutions. The National University of Singapore is at the top for its individual publications (224), which might be due to the establishment of the “Membrane Science and Technology Consortium (MSTC)” in the Department of Chemical & Biomolecular Engineering. Similarly, Tianjin Polytechnic University, also known as Tiangong University (TGU) of China, is at second position with 141 individual publications, and the reason might be due to the establishment of “The National Centre for International Joint Research on Membrane Science and Technology”, which was built by the Ministry of Science and Technology of China in 2016. The history of this centre can be traced back to 1974, when the first industrialized hollow fiber membrane modules were developed. The center has acquired technical equipment and development conditions, such as the research and development of new-type hollow fiber membranes, and the process development of the membranes with a high-level research team of membrane science and technology. Universiti Teknologi Malaysia is at third place with 117 publications, due to the upgradation of its Membrane Research Unit into the “Advanced Membrane Technology Research Centre (AMTEC)” as one of the Centres of Excellence (CoE) in UTM in the year 2008, and is now becoming the Higher Institution Centre of Excellence (HICoE or national CoE), which is recognized by the Ministry of Higher Education Malaysia since November 2015. AMTEC is working extensively on many research projects related to ceramic and polymeric HFM membrane development and applications in water treatment and gas separation.

### 3.4. Prominent Authors

Table 3 lists the 10 most prolific authors in HFM, affiliated to seven different countries. The affiliations of the authors with different technical and engineering institutions and departments showed that HFM research fields are related to water research, the environment and chemical engineering.

T.S. Chung from the National University of Singapore is at the top of the list, having 218 publications in total since 1997. A High document h-index (68) and 13,448 citations of T.S. Chung show his novel and widely accepted research work in HFM. The second author from Universiti Teknologi Malaysia is A.F. Ismail, having a total count of 175 published articles (first article in 1997) with 4578 citations and an h-index of 39. Although the author Li Kang from Imperial College London, UK is third in the list with 125 publications, he received a higher number of citations (6807) compared to the second author, which may be due to the quality of research work or open access publications and the three years of early career start-up i.e., 1994 to 1997. Two authors each from Singapore and Japan are among the top ten, which reveals that their respective institutions have higher contributions in this research area. VOSviewer analysis also confirmed the data of authors as shown in Figure 6 for the density map: a more highlighted area in yellow shows stronger co-authorship among these authors for their research papers and a bolder name shows the most productive authors in HFM research. Figure 7 shows the average publication year of authors, which is determined by the sum of years of all publications of an author divided by the total number of publications. A more recent average publication year of an author depicts his higher contributions more recently. A Larger size and reddish colour of the ball shows more and recent author contributions in an average publication year. Similarly, higher average citations means the sum of all citations of all publications divided by total publications, and a higher value means that the author has been cited more than other authors, as shown in Figure 8.

### 3.5. Analysis of Time-Frequency of Author Keywords

A total of 9027 author keywords were recorded as appeared in HFM research, among which 6654 (73.7%) were used only once, 1061 keywords (11.8%) were used twice and 444 (4.9%) were used thrice. After re-labelling synonymic single words and congeneric phrases of these total keywords that appeared in the title and abstract of all documents in the Scopus database, they were re-labelled for synonymic single words and congeneric phrases, and finally 515 keywords that occurred for more than five times were enrolled in the final analysis for the mapping in VOSviewer.

#### 3.5.1. Terminology and Concept

VOSviewer transformed the data into a visual form and classified the frequently occurring keywords into clusters in the network visualization mode [51]; the results are shown in Figure 9. Larger frames and bolder map labels represent a greater value and impact. Keywords with similar colors belong to the same cluster as mentioned in Figure 5 [34].

#### 3.5.2. Clusters Analysis and Topics of Interest

In our study, software compiled the data of keywords into 17 clusters, as shown in Figure 9. Results showed that “hollow fiber membrane” (HFM) was the most frequently encountered keyword, with 984 occurrences and 396 links to other keywords, followed by “hollow fibers” with 670 occurrences and 330 links, “membrane contactor” with 285 occurrences and 174 links and ultrafiltration, gas separation, PVDF, MBR and so on, as shown in Figure 9. The keyword “HFM”, which appeared the most (total link strength 1833), and “hollow fibers” (total link strength 1437) had strongest links with “ultrafiltration”, “membrane contactor”, “membrane fouling” and “gas separation” based on visualization analysis shown in Figure 9. The closer the keyword is to HFM, a more prominent appearance and a bigger size of frame show the stronger link strength of keywords with HFM and its wider use in most HFM related papers.

It can also be observed from the visualization map that HFM has also seen co-occurrence with conceptual keywords including “surface modelling”, “mass transfer”, “filtration”, and “permeability”. Each keyword has its own research-based origin and has a connection with a particular field. It is interesting to note that the top 20 keywords listed as per their decreasing total link strength (1437–183), which appeared most frequently in the bibliographic mapping, are related to two major applications of HFM i.e., gas separations and water treatment.

The 17 clusters are differentiated into different colors, as shown in Figure 9. These clusters confirm the research streams obtained from the bibliographic coupling of 5626 articles. The main cluster is cluster one, with 53 items (keywords), marked in red, mostly related with water treatment and problems such as membrane fouling and biofouling, which appeared in a total of 239 occurrences, with 174 links and a link strength of 456 along with less occurred key words of “fouling control” and “fouling mechanism”. Furthermore, the most prominent keywords include “membrane bioreactor” (MBR), (188 occurrences, 109 links), “ultrafiltration” (226 occurrences, 143 links) and “microfiltration” with 102 occurrences and 85 links with other keywords. Other terms, such as “water reuse”, “treatment”, “textile water”, “oily wastewater” and “municipal wastewater”, are all linked with HFM and MBR, ultrafiltration and microfiltration, which are prominent for water treatment technologies. Hollow fiber MBR technology is applicable to many sectors, including municipal, industrial and water treatment. The use of an MBR process for water reclamation can reduce the demand for potable quality water on local supplies and can reduce pollution from waste discharges into local water bodies.

It has been shown that in cluster two there are 41 keywords, and the most frequently occurring keywords include “gas separation” (194 occurrences, 136 links and 425 TLS) and “dual-layer HFM” (48 occurrences, 53 links and TLS of 88). Keywords, such as “separation”, “CO_2_/N_2_ separation”, “SO_2_”, “natural gas purification” and “methane”, along with “polyimide”, “polyetherimide” and “PVDF”, also appeared in close relation with HFM and gas separation keywords, showing a strong inter-relationship for applications and the fabrication of HFM. In gas separation, “methane” is most closely linked, along with keywords “polyimide”, “polyetherimide” and “polybenzimidazole composite membranes” used in HFMs for gas separation.

In cluster three, there are 41 items, and major areas include applications in biochemistry, medical and analysis methods, which is apparent by keywords such as “adsorption” (49 occurrences, 53 links and TLS 95), “carbonic anhydrase”, “enzyme immobilization”, “protein adsorption”, “lipase”, “HPLC”, “gas chromatography” and numerical analysis. All of these terms are closely linked with “HFM”, “hollow fibers”, “mixed matrix membrane”, “ion exchange”, “ceramic HFMs” and “ultrafiltration”.

Cluster four include 41 items, with “PVDF” being the prominent word with maximum occurrences (159), links (123) and TLS (377), followed by the keyword “membrane distillation” with 106 occurrences, 110 links and 259 TLS. It has been found that in this cluster, the word interconnected most closely with HFM is “membrane distillation”, which is further interlinked with “PVDF”, “TiO_2_”, “desalination”, “seawater”, “CO_2_ separation”, “composite membranes”, “hydrophilic membranes”, “direct contact membrane distillation (DCMD)”, “PTFE” and “membrane contactor”. It validates the idea that PVDF-based composite hollow fiber membranes having hydrophilic properties can be used effectively in, but are not limited to, seawater desalination and gas separation applications. The use of PVDF in HFM fabrication and its multiple applications has been reported by [6].

Cluster five contains 38 items, and the keyword that appeared most frequently in this cluster is “wastewater”, which appeared 82 times in publications, with 85 links and TLS of 179. The closely interconnected keywords with HFM and wastewater include “MBR”, “biodegradation”, “phenol”, “activated carbon”, “biological treatment”, “microfiltration”, and “volatile organic carbons” (VOC). It can be predicted that this cluster deals with the use of activated carbon for wastewater treatment along with the use of HFM-based MBR for the biodegradation of VOC, such as phenol, in order to clean the water.

Cluster six is mainly concerned with the fabrication, properties, drawbacks, modifications and applications of HFM, as shown in Figure 9, for the overlay visualization of keywords connected with HFM. By further zooming the cluster for HFM, all closely related keywords can be seen in Figure 10. For this data, it has been observed that HFM is the most frequently occurring keyword overall with 984 occurrences, 396 links and a TLS of 1833. The most closely found keywords with a strong TLS appeared to be “antifouling”, “composite membranes”, “graphene oxide”, “computational fluid dynamics”, “forward osmosis” and many more. For problems of membrane fouling, the closely related terms, such as “polyethersulfone” (PES), “polyvinyl alcohol” (PVA), “hydrophilicity” and “pressure retarded osmosis”, show their role in minimizing the membrane fouling. Furthermore, for membrane fabrication, terms such as “composite membranes”, “PVDF”, “silver nanoparticles”, “spinning”, “polyvinylpyrrolidone” (PVP), “silica”, “polyacrylamide”, “PES” and “polysulphone” (PS) were found to be in close relationships. These relationships have also been reported by many authors [52,53,54], which also supports our assumption that these materials play a vital role in HFM synthesis. From the applications side, the keywords most frequently and closely found are “water treatment”, “dye removal”, “methylene blue”, “gas separation”, “CO_2_ separation” and “haemodialysis”, as shown in Figure 10. If we go for HFM modification, then it can be found that terms such as “surface modification”, “PES”, “hydrophilicity” and “pore size distribution” appear most frequently and are closely related to HFM, which is also reported in the literature for the better performance of HFM with respect to biofouling. Keyword “HFM” shows a maximum link strength of 53 with keywords “PVDF” and “gas separation”, followed by “ultrafiltration” (50), “PES” (39), “membrane contactor” (35), “PS” (34), “MBR” (30), “pervaporation” (27), “nanofiltration” (26), “thermally induced phase separation” (25), “water treatment” (17), “wastewater” (14) and “CO_2_ separation” (13). The progressive decrease in link strength of HFM with the keywords mentioned above indicates the decreasing order of research interest of HFM in those topics. It can also be concluded that the maximum research interest for HFM is related to the use of PVDF in its fabrication and applications in gas separations. It can be found that “graphene oxide”, with 20 occurrences, 34 links and 50 TLS, is interlinked with keywords such as “HFM”, “hollow fibers”, “mixed matrix membrane”, “CO_2_ separation” and “nanofiltration”, and also a weak link with “artificial kidney”, as shown in Figure 11.

All 17 clusters can be tracked by interested readers using the URL link http://bit.ly/3bRwpFx accessed on 7 August 2021, which is available online for more details and to explore new emerging research areas. However, it is worth mentioning cluster 10, which includes the biomedical application area of HFM and hollow fibers. This cluster consists of mostly biological and medical terms. Terms such as “artificial kidney”, “artificial lungs”, “bioartificial liver”, “haemodialysis”, “biocompatibility”, “membrane oxygenator”, “life support” and “paediatrics” appeared the most frequently. It has been found that composite HFM (mixed matrix membrane) has a strong link strength with these keywords. Due to the presence of a large number of pores in HFM and use of biocompatible materials, such as PES and cellulose, they found applications in blood oxygenators for cardiopulmonary bypass in paediatrics as both a life support and artificial lungs, which is also supported by the work published by many authors [55,56,57,58].

#### 3.5.3. Research Hotspots and Future Trends

Figure 12 demonstrates the network map for overlay visualization based on the average publication year of the emerging topics according to the keywords used, covering the data period from January 1970 to December 2020. The colour scheme used indicates the current publications from red to purple in decreasing order. We have focused on this section for keywords that mostly occur in recent average publication years. From Figure 12, it can be observed that, in recent publication years, mostly occurring keywords address the main problems associated with membranes, such as energy consumption, irreversible fouling, CO_2_ resistance, the industrial scale use of HFM and stability issues. Words such as “antifouling”, “solar energy”, “desalination”, “Computational Fluid Dynamics” (CFD), “process simulation” and “heat & mass transfer” are linked with the issues of HFM. Some potential emerging topics related to main keywords such as “HFM”, “hollow fibers”, “gas separation”, “water treatment”, “membrane contactor”, “membrane fouling” and “MBR” have been listed in Table 4. The contents of Table 4 demonstrate that the most recent and emerging trends in HFM research areas from the present work are in accordance with the research trends reported in various HFM-related literature reviews by various authors [5,59,60,61].

The membrane types and processes that appeared most frequently in recent years include mixed matrix membranes, ceramic membranes, sweeping gas membranes, membrane distillation, forward osmosis, DCMD, photocatalytic degradation, pressure retarded osmosis, hydrophobic modification, dual-layer HFM, superhydrophobic, zeolite membranes, nickel membranes and air gap distillation. HFM materials which appeared frequently in recent average publications years include polydimethylsiloxane, PTFE, PVS, kaolin and graphene oxide, agreeing with those mentioned by [62].

Most recent applications of HFM, as per scientometric analysis of most occurring key words in recent publications, include municipal wastewater, desalination, oily wastewater, dye removal, biogas, CO_2_ capture, CO_2_ separation, arsenic removal, hydrogen production, water vapor separation, organic solvent nanofiltration, syngas and dual layers. From the analysis of the density visualization map as shown in Figure 13, six major research areas have been identified.

Based on the number of articles and author keyword occurrences, there were positive correlations found between the outputs of sub-theme search (applications of HFM) and the central theme (HFM) search. For the three main selected applications of HFM, namely, gas, water and biomedical, each term was individually searched in Scopus using all related terms. For example, the use of gas separation for the key string (search phrase: TITLE-ABS (“hollow fiber membrane”) OR TITLE-ABS (“Dual-layer hollow fiber membrane”) AND TITLE-ABS (“gas separation” OR “gas permeation” OR “gas absorption” OR hydrocarbons OR CO_2_ OR ch4 OR n2 OR ammonia OR h2s OR air) AND (LIMIT-TO (SRCTYPE, “j”)) AND (LIMIT-TO (DOCTYPE, “ar”)) AND (EXCLUDE (PUBYEAR, 2021)) AND (LIMIT-TO (LANGUAGE, “English”)) resulted in a maximum number of research articles (1774) from year 1970–2020 in Scopus, as shown in Figure 14, and the word “gas separation” has appeared 194 times in VOSviewer (Figure 15). However, if we combine the occurrences of all gas-related keywords, such as “CO_2_”, “CO_2_ separation”, “CO_2_ capture” and other gas related keywords, the total occurrences become >500, which also supports the Scopus data. It was followed by “wastewater treatment” (1294 articles and >400 occurrences) and medical applications such as “blood oxygenator”, “hemodialysis”, “artificial liver”, “artificial kidney”, “biocompatibility” and similar (509 articles and >150 occurrences).

Furthermore, research interest in certain areas can also be analysed by the link strength of two keywords. It has been found that gas separations mostly deal with “CH_4_” “H_2_S”, “ammonia” and “CO_2_”, and the membrane contactor and HFM are also found to be strongly linked with these keywords, showing HFM’s application in gas separations as reported by [63]. Water treatment, desalination, denitrification and wastewater are closely linked with the keywords “membrane distillation”, “microfiltration”, “ceramic membranes”, “PVDF” and “nanofiltration”, which is also in accordance with the report by Kim et al. [39]. Similarly, for medical applications, keywords “haemodialysis”, “artificial kidney” and “artificial liver” are closely linked with hollow fibers, HFM, PES and mixed matrix membrane and membrane oxygenators [64,65]. It is interesting to note that the gas separation mechanism of HFM works on the same principle as in the medical applications, such as artificial lungs and membrane oxygenators, for oxygen in which CO_2_ permeation and separation occurs. Therefore, gas separation applications are mostly linked with the medical field. Similarly, nanofiltration technology used for water treatment in HFM has also been applied in the medical field in artificial kidney and haemodialysis for blood purification [66].

It has also been found that most of the publications that were related to the gas applications and wastewater treatment came from China, whereas for biomedical applications, USA is leading, as shown by Figure 16. For gas separation, Singapore, Iran and Malaysia are among the top five countries, which is in accordance with the bibliographic analysis for the countries. However, for water treatment applications of HFM, China is followed by Singapore, USA, South Korea and Japan. It is interesting to note that for more sophisticated applications of HFM in the biomedical field, a different trend has been observed, where Japan is second and two European countries, Germany and the Netherlands, also make their entry in the list of top five, whereas China is number four. In general China, USA and Japan are the most contributing countries in the field of HFM development and applications.

## 4. Limitation of Study

By restricting the keywords “hollow fiber membrane” and “dual-layer hollow fiber membrane” in the title and abstract, the search results may not cover all HFM-related studies available on Scopus. This is because some researchers may have used other similar terms instead e.g., “porous membrane”, “hollow fiber”, “polymeric hollow fiber membrane”, “hollow fibre membrane”, “hollow fiber mixed matrix membrane” and similar terms. In addition, by restricting the search to only “articles” from total of 7363 retrieved documents, we have come up to 5626 articles based on title and abstract analysis (full body of a paper if required) but still there is possibility of selecting some review papers in the list of 5626 scientific papers due to the use of a term other than “review” in the selected articles list.

## 5. Conclusions and Recommendations

With the increase in the global population, problems directly linked with the necessities of a common man (e.g., drinking water) are also increasing, which has indirectly resulted in more and more academic papers that focus on that research area. It is especially important to evaluate the quality of such a huge number of research papers and obtain valuable information that can help to find solutions to these issues. The present study surveyed the research on HFM from the very first paper indexed in the Scopus database in the year 1970 until the end of the year 2020. Relevant information related to annual publication distribution, journals, main countries and institutions was analyzed. Based on 5626 articles published in 159 journals from 82 countries (including 30 papers from undefined source titles), the Scopus database and bibliometric analyses (VOSviewer software) were used to obtain important information on the research trends and main applications of HFM. Scientometric analysis was found to be an innovative strategy for figuring out useful points quickly and exactly from massive information. Since the article on HFM first appeared in 1970, the number has grown continuously until it broke through 100 total articles in the year 1985. Moreover, for a single year of 2000, it crossed the limit of 100 articles, and the overall trend has been on the rise until 2020. It was noted that 32% of papers were categorized under chemical engineering, 21.5% under chemistry, 14% under biochemistry and medicine and 8.9% under environmental sciences. From the overall analysis, gas separation was the most studied application, followed by water treatment and biomedical applications. The *Journal of Membrane Science* was the top listed journal with 1167 documents, followed by *Desalination* with 314 papers. China is the most productive country; it has the most productive institutes and funding agencies, and Tianjin Polytechnic University of China was the top institution with the highest number of publications. In this way, China showed to be a dominant contributor to gas separation and water treatment-related applications. Research on HFM shows an increasing trend, with >300 per year publications in the last five years. Thus, the research trends presented in this study will lead researchers to establish future directions in new studies for variety of applications. Due to its effectiveness, it is highly recommended to apply the bibliometric analysis to those key words that are related to membrane material, membrane design, membrane characterization, membrane process design, membrane system design etc.

The use of hollow fibers in membrane separation has become one of the emerging technologies that underwent a precipitous growth during the last few years. Based on the current analysis, it has been found that challenges, such as lowering the operational and capital cost, energy demand and the development of a more efficient system with new advanced materials, such as new polymers that have the best antifouling properties, should be addressed in the future. The use of HFM in gas separation, municipal and industrial wastewater treatment, and the biomedical field, with more efficient fouling control to guarantee the long-term performance of the system, are trending research areas. The application of integrated software technologies to design and assemble modules using new polymers and ceramics with hybrid systems and fine control on rejections are the hotspots in this field. Similarly, forward osmosis with better antifouling properties, micro and ultrafiltration systems, HF-MBR, mixed matrix membranes and hydrophobic HFM contactors are trending systems to be further explored in the future.

Bibliometric analysis can be performed using multiple data sources that are useful for a wider range of studies, such as Web of Science, US patents and Google Scholar. An advantage of using Web of Science search results is that it automatically shows the most popular articles in the field with a feature called "hot paper" that Scopus still lacks. This hot paper feature displays major articles that are recognized shortly after publication and are reflected in a rapid and significant number of citations. It is also recommended to compare the outputs from two or more databases using other software which can give a more comprehensive analysis of the work performed in various research fields of HFM. This type of study can be extended further for other research areas in the membrane technology in order to obtain an overall quick and comprehensive overview and to find the emerging trends in that area. More focused research is still awaited in the field of HFM in alignment with the future needs, specifically in biomedical applications. The use of new advanced and inexpensive materials and the latest fabrication technologies are also required to overcome the issues of stability, selectivity, permeability, and fouling. This piece of work will be helpful to draw the attention of researchers, policy makers and individuals to understand the research trends in HFM and to uncover the potentials and opportunities for future research.

## Figures and Tables

**Figure 1 membranes-11-00600-f001:**
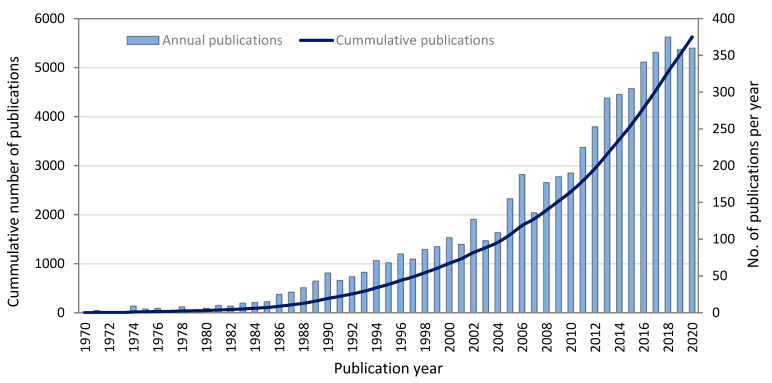
The annual and cumulative distribution of Scopus-indexed research articles on hollow-fiber membranes from 1970 to 2020.

**Figure 2 membranes-11-00600-f002:**
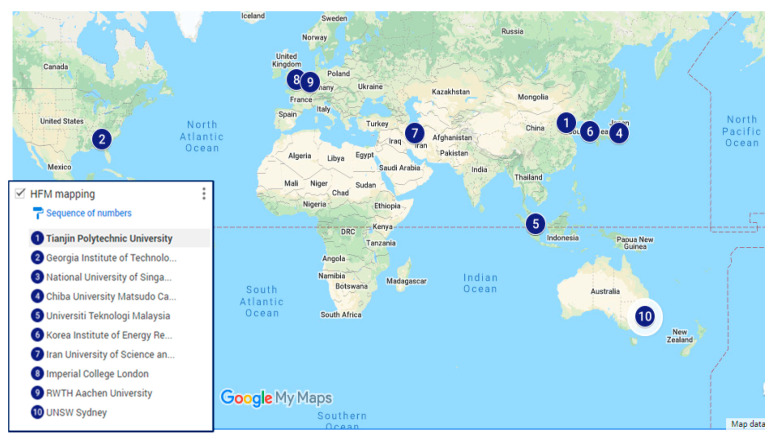
The top 10 most productive countries and academic institutions in HMF publications. For ref, please visit: https://www.google.com/maps/d/u/0/edit?mid=18w6RI2rrr67KIdfRM85rBKG4idoYqh6N&ll=2.367478565105216%2C0&z=2 accessed on 7 August 2021.

**Figure 3 membranes-11-00600-f003:**
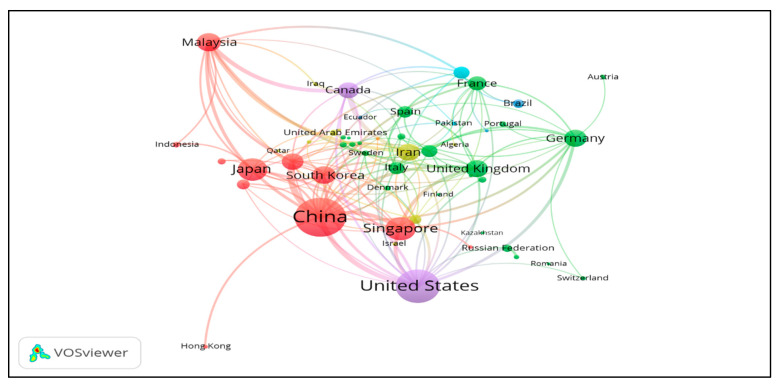
A screenshot of bibliometric map created based on co-authorships with network visualization mode can be accessed online at: http://bit.ly/38TXSom accessed on 7 August 2021.

**Figure 4 membranes-11-00600-f004:**
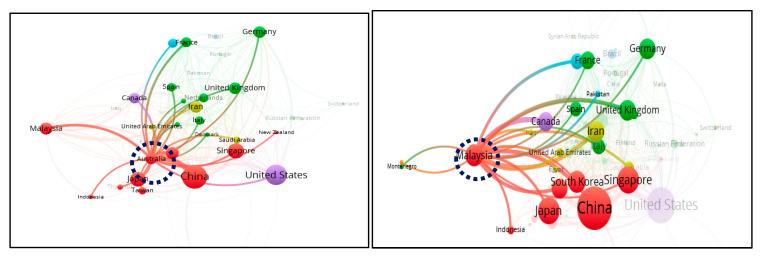
Links of Australia and Malaysia showing inter-country collaborations of HFM documents.

**Figure 5 membranes-11-00600-f005:**
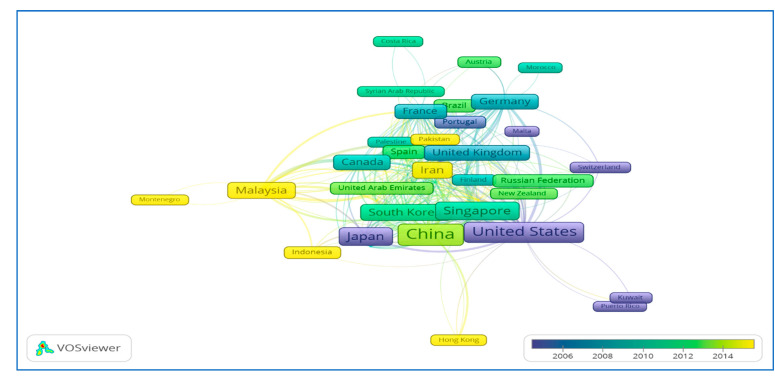
A screenshot of the bibliometric map created based on co-authorships with overlay visualization mode. Available online at http://bit.ly/38TXSom accessed on 7 August 2021.

**Figure 6 membranes-11-00600-f006:**
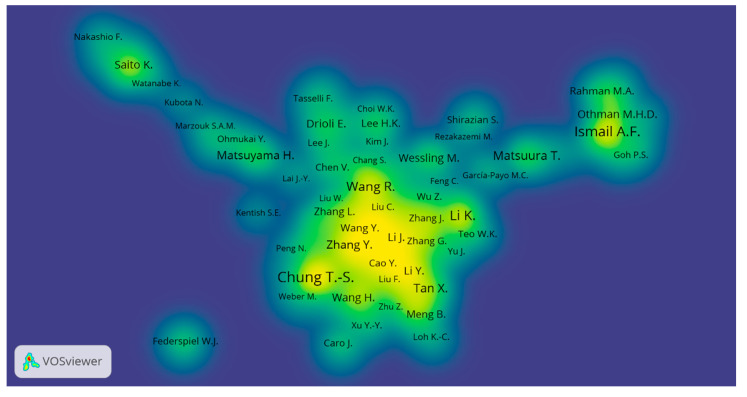
Density map showing the authors inter-relationships for co-authorship. Available online at http://bit.ly/2KoH5Aj accessed on 7 August 2021.

**Figure 7 membranes-11-00600-f007:**
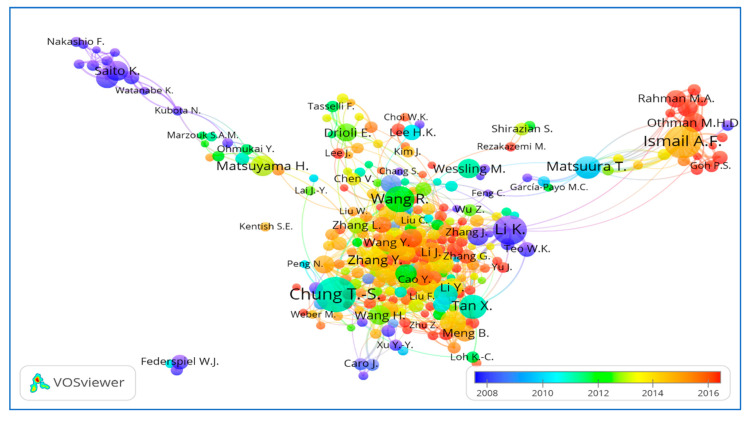
Overlay visualization of collaborative network between authors, showing the average publication year of authors. Available online at http://bit.ly/2KoH5Aj accessed on 7 August 2021.

**Figure 8 membranes-11-00600-f008:**
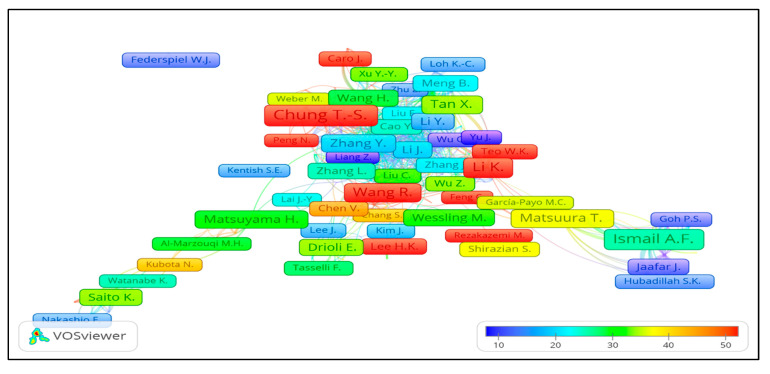
Overlay visualization showing the author number of average citations per year as shown by given color scale. For online view of data in VOSviewer http://bit.ly/2KoH5Aj accessed on 7 August 2021.

**Figure 9 membranes-11-00600-f009:**
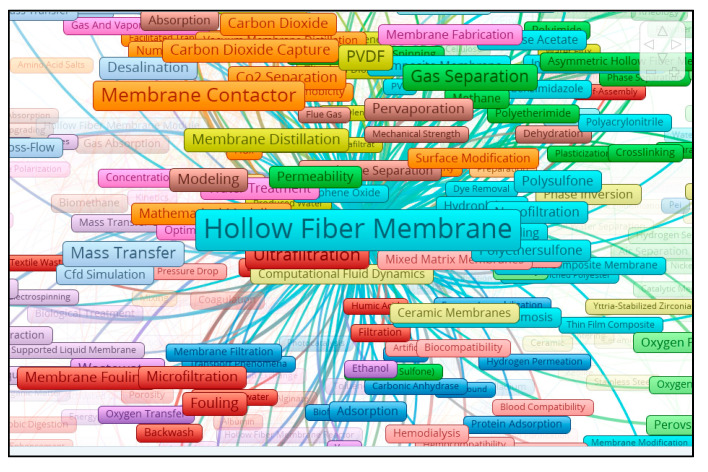
A screenshot of the bibliometric map created based on author keywords co-occurrence with overlay visualization mode. Available online at: http://bit.ly/3bRwpFx accessed on 7 August 2021.

**Figure 10 membranes-11-00600-f010:**
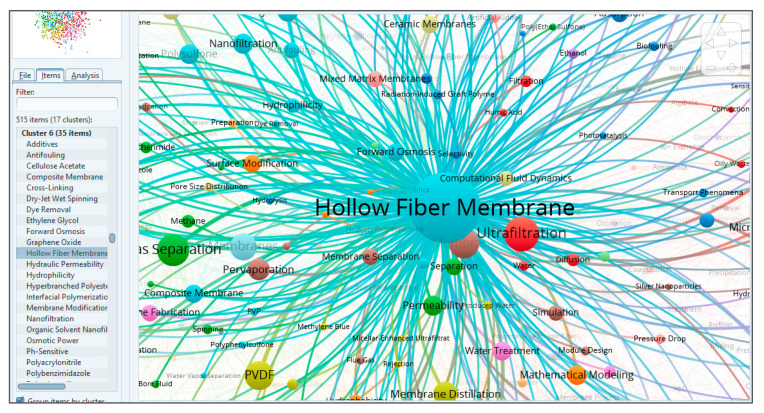
Screenshot of cluster six, having 35 items, showing links of HFM with other keywords. Available online at http://bit.ly/3bRwpFx accessed on 7 August 2021.

**Figure 11 membranes-11-00600-f011:**
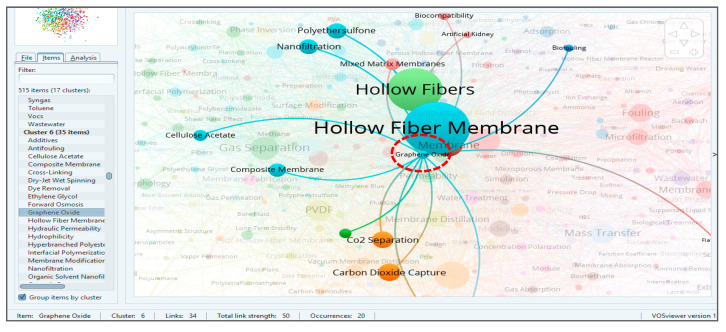
Screenshot of keyword graphene oxide in cluster six showing the connection with other connected research areas.

**Figure 12 membranes-11-00600-f012:**
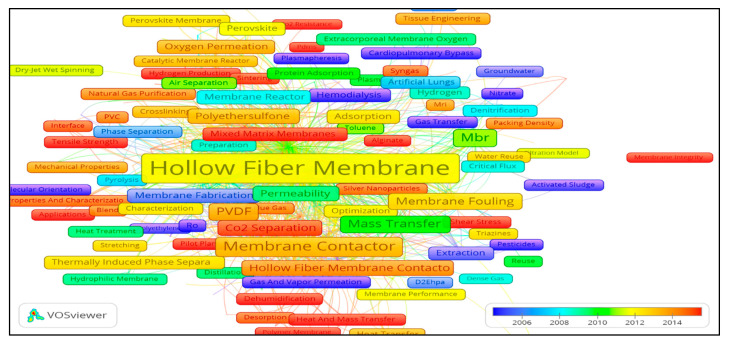
Overlay visulization map of keywords based on author keywords co-occurrence. Can be viewed online at http://bit.ly/3bRwpFx accessed on 7 August 2021.

**Figure 13 membranes-11-00600-f013:**
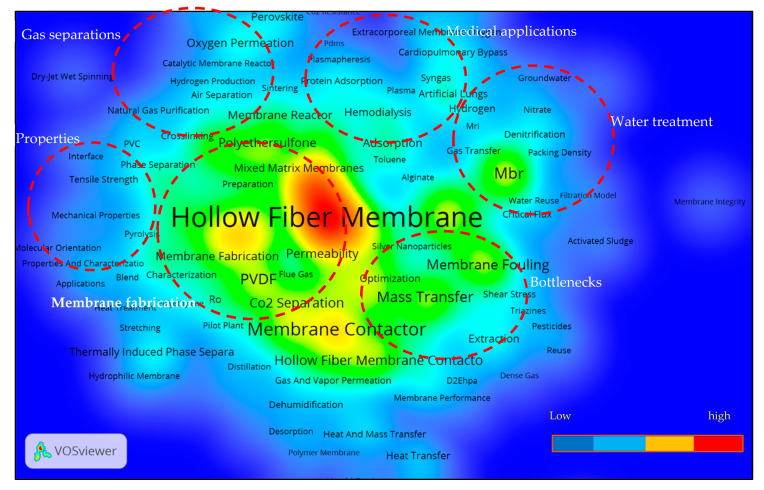
Keyword map using density visualization showing the key research areas of HFM research. Available online at http://bit.ly/3bRwpFx accessed on 7 August 2021.

**Figure 14 membranes-11-00600-f014:**
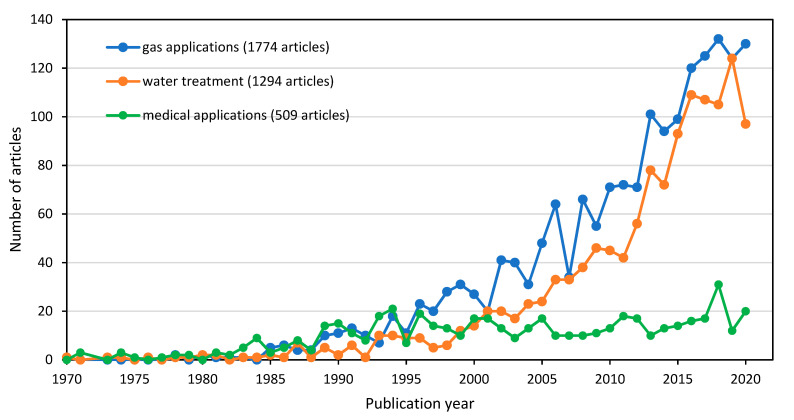
Research trends of the selected major applications in HFM using Scopus database.

**Figure 15 membranes-11-00600-f015:**
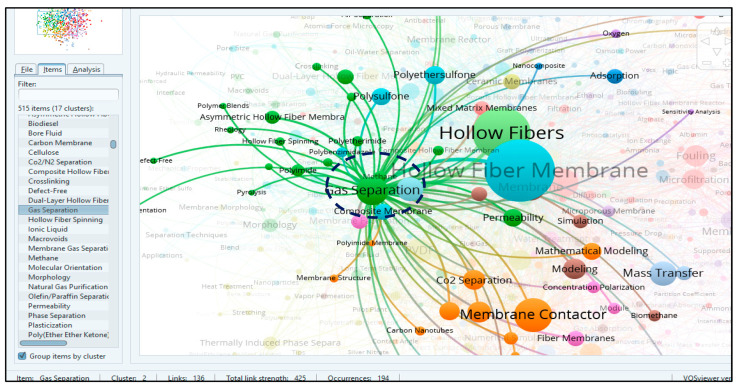
Screenshot of network visualization of keyword “gas separation” showing 194 occurrences. Can be accessed online at http://bit.ly/3bRwpFx accessed on 7 August 2021.

**Figure 16 membranes-11-00600-f016:**
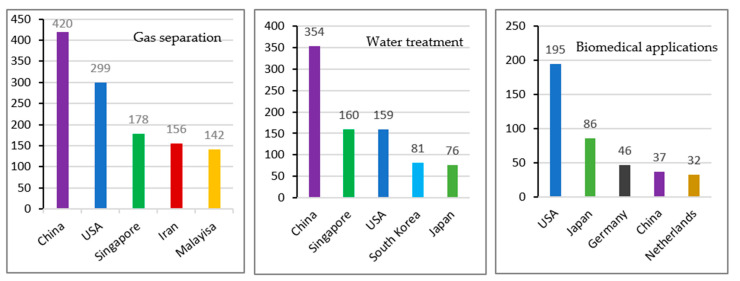
Top five countries with the highest publications on the selected HFM major applications.

**Table 1 membranes-11-00600-t001:** The top 10 most productive journals based on total number of articles.

Rank	Journal Name (IF)	^1^ TP	TP %	^2^ TC	Most Cited Article	Times Cited	Publisher
1	Journal of Membrane Science (7.158)	1167	20.74%	50,776	Characterization of novel forward osmosis hollow fiber membranes [41]	428	Elsevier
2	Desalination (7.248)	315	5.60%	9125	Fouling control in activated sludge submerged hollow fiber membrane bioreactors [42]	241	Elsevier
3	Separation and Purification Technology (5.257)	282	5.01%	7625	Influence of membrane wetting on CO_2_ capture in microporous hollow fiber membrane contactors [43]	292	Elsevier
4	Industrial & Engineering Chemistry Research (3.684)	140	2.49%	4392	Gas absorption studies in microporous hollow fiber membrane modules [44]	281	American Chemical Society
5	Journal of Applied Polymer Science (2.257)	139	2.47%	2421	Effect of polyvinylpyrrolidone molecular weights on morphology, oil/water separation and mechanical and thermal properties of polyetherimide/polyvinylpyrrolidone hollow fiber membranes [45]	106	Wiley-Blackwell
6	Desalination and Water Treatment (1.324)	121	2.15%	820	Vacuum membrane distillation for an integrated seawater desalination process [14]	65	Desalination Publications
7	AICHE Journal (3.625)	103	1.83%	3272	Dispersion-free solvent extraction with microporous hollow-fiber modules [40]	468	Wiley-Blackwell
8	Chemical Engineering Journal (9.43)	86	1.53%	2212	CFD simulation of natural gas sweetening in a gas–liquid hollow-fiber membrane contactor [46]	149	Elsevier
9	Chemical Engineering Science (3.78)	79	1.40%	3074	Hydrophobic PVDF hollow fiber membranes with narrow pore size distribution and ultra-thin skin for the freshwater production through membrane distillation [47]	197	Elsevier
10	Journal of Chromatography A (3.861)	77	1.37%	3218	Determination of organochlorine pesticides in seawater using liquid-phase hollow fibre membrane microextraction and gas chromatography-mass spectrometry [48]	134	Elsevier

^1^ TP (total publications), ^2^ TC (total citations).

**Table 2 membranes-11-00600-t002:** The top ten most active countries and institutions of HFM publications.

Rank	Country/Territory	^1^ TPC	TPC%	SCP	SCP %	Most Productive Academic Institution	TIP	QS Ranking 2021
1	China	1260	22.40	890	70.63	Tianjin Polytechnic University	141	387
2	United States	945	16.80	680	71.96	Georgia Institute of Technology	64	80
3	Singapore	487	8.66	306	62.83	National University of Singapore	224	11
4	Japan	464	8.25	380	81.90	Chiba University	46	488
5	Malaysia	306	5.44	158	51.63	Universiti Teknologi Malaysia	117	187
6	South Korea	297	5.28	205	69.02	Korea Institute of Energy Research	35	39
7	Iran	282	5.01	186	65.96	Iran University of Science and Technology	31	601–650
8	United Kingdom	271	4.82	133	49.08	Imperial College London	50	8
9	Germany	269	4.78	150	55.76	Rheinisch-Westfälische Technische Hochschule Aachen	32	145
10	Australia	257	4.57	87	33.85	University of New South Wales	36	44

^1^ TPC (total publication count), SCP (single country publications), TIP (total individual publication of institute).

**Table 3 membranes-11-00600-t003:** List of the 10 most prolific authors in HMF research area.

Rank	Author	Scopus Author ID	TP ^1^	Year of 1st Publication	Document h-Index in the Specific Area of Hollow Fiber Membrane *	TC ^2^	Current Affiliation	Country
1	Chung, T.S.	7401571059	218	1997	68	13,448	National University of Singapore	Singapore
2	A.F., Ismail	7201548542	175	1997	39	4578	Universiti Teknologi Malaysia	Malaysia
3	Li, Kang	6505451560	125	1994	51	6807	Imperial College London	United Kingdom
4	Wang, Rong	35081334000	100	2000	47	6191	Nanyang Environment & Water Research Institute	Singapore
5	Tan, Xiaoyao	7202120957	91	2000	31	3048	Tianjin Polytechnic University	China
6	Matsuura, Takeshi	36048717100	84	1991	33	3013	University of Ottawa	Canada
7	Liu, Shaomin	35242760200	79	2001	26	2409	Curtin University,	Australia
8	Matsuyama, Hideto	57201543303	70	2003	26	2043	Kobe University	Japan
9	Koros, William J.	7102165550	69	1990	30	3183	Georgia Institute of Technology	United States
10	Saito, Kyouichi	7406511537	65	1988	30	2163	Chiba University	Japan

^1^ Total publications (TP), ^2^ Total citations (TC); * Document h-index and total citation (TC) are different than the overall h-index.

**Table 4 membranes-11-00600-t004:** Recent flashpoints in HFMs’ development and applications.

No.	Keyword	Average Publication Year	No. of Occurrences
1	Sweeping gas membrane distillation	2018.44	23
2	Graphene oxide	2018.40	20
3	CO_2_ resistance	2018.20	5
4	NIPS (Non-solvent induced phase separation)	2018.17	6
5	Nanofluid	2018.00	6
6	Nickel membrane	2018.00	5
7	Superhydrophobic	2017.78	9
8	Anaerobic membrane bioreactor	2017.75	8
9	Zeolite membrane	2017.70	11
10	Potassium carbonate	2017.60	11
11	Dye removal	2017.60	10
12	Water vapor separation	2017.40	14
13	Photocatalytic degradation	2017.33	9
14	Electrospinning	2017.29	7
15	Perovskite oxide	2017.20	13
16	Polydopamine	2017.14	7
17	Hydrogen production	2016.71	7
18	Antifouling	2016.50	42
19	Oil—water separation	2016.38	21
20	Irreversible fouling	2016.17	6

## Data Availability

Not applicable.

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
