# Peer review of "Research and Development Journey and Future Trends of Hollow Fiber Membranes for Purification Applications (1970–2020): A Bibliometric Analysis"

_membranes, 2021, doi:10.3390/membranes11080600_

Round 1
Reviewer 1 Report
Article entitled „Research and development journey and future trends of hollow fiber membranes for purification applications (1970-2020): A bibliometric analysis” is an interesting peace of work. Authors presented a bibliometric and scientometric analysis of publication related to hollow fiber membrane (HFM) over last 40 years. However, I do have problem to classify the manuscript to either research article (article) or review, since it doesn’t present any new research but present an interesting overview of research history, research centers, the most influencing authors, etc. aiming to show the future directions in publications (not necessarily in research) and possible “hot topics”. Therefore, in my opinion, authors should clearly justify why this manuscript should be treated as research article.
Nevertheless, before taking the final decision about acceptance for publication of it, authors should consider the following points.
Major points:
- Authors should deliver in additional/supportive materials the input .csv files, to ensure the transparency of their research and used data.
- Line 523. Authors differentiated 17 clusters, could clearly specify which are those? While reading it is not clear.
- Table 4. What is the exact meaning of numbers representing the average publication year?
- Line 713. Germany and Netherlands are not new European countries.
- The presented conclusions are more like summary. Could authors elaborate or clearly state what is/are future trend/s in HFM? The future trends should be clearly stated since it highlighted in the title.
- Could authors assess the industrial and/or real applicability of HFMs based on the performed bibliometric analysis? How often such keywords as design, simulation, modelling, mass transfer are found in relation to HFM?
Minor points:
- Line 290 and other places. Full name is “AICHE Journal”, please make sure to use the proper name everywhere.
- Line 562. Too many spaces before “[6]”. I’ve the feeling, that too many spaces are used in many places.
- Line 697. Do not mix the citation styles.
- Line 726 and 722. Duplication.
Reviewer 2 Report
The authors conducted a bibliometric study on the research trend on hollow fiber membranes over the past few decades. The content is interesting to readers and fits well in the special issue. Just two comments below:
- The authors screened papers using two strings "hollow fiber membrane" and "dual-layer hollow fiber membrane", and as they mentioned in Section 4, relevant papers can be titled with alternative terms - was there a reason that no more than two strings were used for the searching? Are the authors positive that the majority of papers have been covered in the study and the results are not biased?
- The authors summarized recent highlights for hollow fiber membrane study in Table 4, which is good. In order to understand how this evolves over time, it's helpful to see this type of information for different time periods, i.e. the authors can present the most popular research topics for each decade and discuss the shift.
Reviewer 3 Report
The manuscript contains some bibliometric analysis. It reads more like a literature search by a student rather than an article. If any, I would recommend to publish it as a review article and not research article. However, I have doubt that that this paper will be read or well-received by the community.
1. The authors need to justify the need for such a manuscript. Who is the targeted audience? The information presented can be simply obtained from web of knowledge if anyone is interested in such data.
2. Did the authors check the validity of the presented data? What methodology was used?
3. There seem to be errors in the manuscript. Table 3 presents specific authors' output. The total citations and h-index among other data in the table does not match with a quick search in GoogleScholar, Scopur or Web or science. For instance, some authors have more than 100 h-index but the table shows otherwise.
4. Table 2 focuses on countries. However, some countries' research in hollow fiber membranes is dominated by a single group; e.g. in UK is listed but almost exclusively all work is coming from the Li group at ICL. It seems misleading to present countries when they are only represented with one group. The number of groups (top 3 major contributors) should be presented in this table and their percentage contribution to give a more accurate picture.
Round 2
Reviewer 3 Report
This reviewer (in line with the other reviewer) fails to see how this work can be categorized as a research article, and who will actually read and build on this 'research'. It seems to be a quick data analysis from Scopus.